# Rethinking Decentralized Learning: Towards More Realistic Evaluations with a Metadata-Agnostic Approach

## Abstract

Decentralized learning has been regarded as a privacy-preserving training paradigm that enables distributed model training without exposing raw data. However, many experimental settings in decentralized learning research assume metadata awareness among participants, which contradicts real-world constraints where participants lack shared metadata knowledge. We distinguish between Metadata-Dependent Supervised Learning (MDSL), which assumes global metadata synchronization, and Metadata-Agnostic Zero-Shot Learning (MAZEL), where participants do not share metadata. Our contributions (1) highlight the difference between MAZEL and MDSL; (2) present empirical evidence demonstrating that long-held claims of MDSL-based decentralized learning may not hold under MAZEL settings; (3) provide benchmarks using up to 8–16 diverse datasets to rigorously evaluate newly proposed decentralized methods under real metadata-agnostic cases; and (4) propose two-stage and cosine gossip schedulers to optimize communication efficiency. Our code is available at: https://anonymous.4open.science/r/More-Realistic-Evaluations.

## 1 Introduction

**Motivating question**: *How can we evaluate decentralized learning algorithms in a way that better reflects real-world constraints, such as the absence of shared metadata and heterogeneous data?*

Decentralized learning offers a promising framework for peer-to-peer collaborative learning across geographically dispersed resources without sharing raw data. In such collaborative scenarios, a collective of agents, often from diverse domains, participate in joint training processes without disclosing sensitive information about their local datasets. This privacy-preserving feature is particularly useful in light of strict data protection regulations. However, we note that there is a discrepancy between how decentralized learning is experimentally evaluated in research and how it is intended to function in real-world deployments.

A notable example of this discrepancy lies in the implicit assumption of "metadata awareness." In many decentralized learning experiments, researchers commonly assume prior knowledge of dataset distributions across the participating agents. For instance, it is common practice to simulate non-IID settings by sampling from CIFAR-100 via a Dirichlet distribution with a specific parameter (e.g., $\alpha = 0.1$) (Yurochkin et al., 2019; Hsu et al., 2019). In such an experimental design, although each node's data distribution is distinct, the overarching metadata, such as the total number of classes, is still treated as shared global information. We refer to this experimental setting as Metadata-Dependent Supervised Learning (MDSL). MDSL implicitly requires that all agents know, for example, the classification categories used at every node.

Contrary to MDSL, many real-world decentralized learning deployments require a more restrictive assumption, i.e., that agents do not share details regarding the nature or scope of their local datasets. Here, privacy constraints or simple lack of knowledge may prevent the system from synchronizing metadata (e.g., the number and type of classes in classification tasks). We designate this experimental setting as Metadata-Agnostic Zero-Shot Learning (MAZEL), where no explicit communication of metadata, such as classes and labels, is required.

Importantly, we note some criticized claims and conclusions about decentralized learning in the current literature under the MDSL experimental setups. E.g. "Local models in decentralized learning

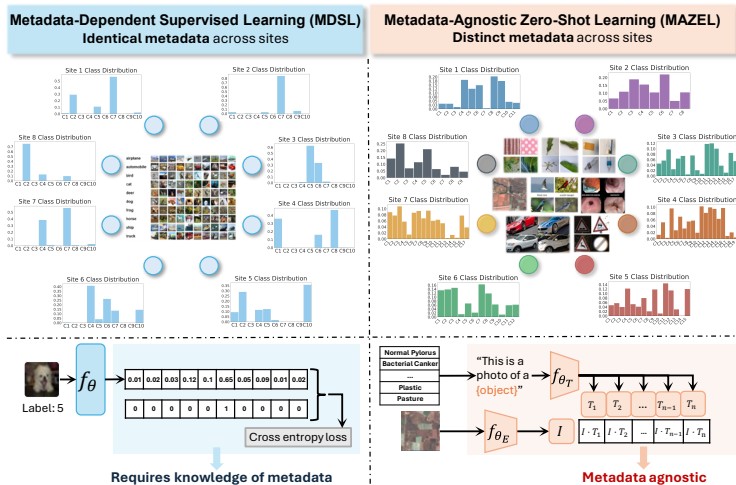

Figure 1: Illustration of the metadata-dependent supervised learning (MDSL) and metadata-agnostic zero-shot learning (MAZEL) settings. **(Left)** Traditional decentralized learning assumes homogeneous, Dirichlet-partitioned datasets (e.g., CIFAR-10/100) with shared metadata (class labels, synchronized task alignment). Coordination relies on pre-agreed classification heads and explicit label synchronization. **(Right)** Proposed real-world scenario: nodes host truly heterogeneous datasets (8–16 distinct domains, e.g., textures, mango leaf, garbage, satellite images, medical imaging, cars, traffic signs) with no shared metadata (unknown classes, divergent label spaces). CLIP-based zero-shot alignment replaces metadata-dependent coordination by leveraging multimodal embeddings for cross-node task alignment. **(Bottom)** Contrast between supervised learning (metadata-dependent cross-entropy loss) and CLIP-driven training (metadata-agnostic alignment).

often generalize poorly to a global test set in highly heterogeneous scenarios (Lin et al., 2021; Vogels et al., 2021)." and "Decentralized learning is known to have much slower convergence compared to centralized learning (Lian et al., 2017; Koloskova et al., 2020)."

In examining scenarios closer to real-world requirements, we find that these claims are **not necessarily correct** for the "gradient-and-gossip" protocols but rather specific to the MDSL experimental settings. Under the MAZEL framework, there is no necessity to share predefined metadata. Preliminary findings suggest that local models can achieve strong performance on the global test set without incurring communication overhead, while effectively balancing both local and global performance.

Our contributions in this paper are: (1) Highlight the limitations of current MDSL-based experimental designs in decentralized learning; (2) introduce more realistic zero-shot testing baselines aligned with the MAZEL experimental settings; (3) demonstrate that several long-held assumptions about decentralized learning performance and communication requirements may not hold under MAZEL; (4) provide benchmarks using 8 and 16 diverse datasets to rigorously evaluate newly proposed decentralized methods under real metadata-agnostic cases; (4) propose two-stage and cosine gossip schedulers for better communication efficiency and to avoid performance oscillation when converges.

## 2 PRELIMINARIES

### 2.1 DECENTRALIZED LEARNING

Decentralized learning allows collaboratively train models without the of control of a central sever. In a decentralized learning framework, the setting can be represented as a connected graph $\mathcal{G} = (\mathcal{V}, \mathcal{E})$, where $\mathcal{V}$ signifies the set of $|\mathcal{V}| = N$ agents involved in the learning process, and $\mathcal{E}$ indicates the communication links among these agents. We also define a mixing matrix between agents using a weighted adjacency matrix $A \in \mathbb{R}^{N \times N}, A_{i,j} \in [0,1] \forall i, j$, where $A_{i,j}$ denotes the strength of the connection from agent $j$ to agent $i$. Each agent $i \in \mathcal{V}$ is characterized by its local model $\theta_i \in \mathbb{R}^d$ and its local data distribution $P_i$. In decentralized learning, two primary settings are considered:

**(1) Personalized Setting** (Vanhaesebrouck et al., 2017; Kharrat et al., 2024)]: This setting focuses on optimizing models for individual agents, where each model is trained to perform well on the agent's local data distribution $P_i$. The corresponding objective is the **Local Population Risk**, which seeks to optimize individual models to perform well on data from their respective local distributions:

$\min_{\{\theta_i \in \mathbb{R}^d\}_{i \in \mathcal{V}}} \left[ F(\boldsymbol{\theta}) \triangleq \frac{1}{N} \sum_{i \in \mathcal{V}} \mathbb{E}_{x_i \sim P_i} l(\theta_i; x_i) \right]$;

**(2) Generic Setting** (Koloskova et al., 2020). This setting aims to train a single consensus model that performs well on the entire data distribution. The corresponding objective is the **Global Population Risk**, which focuses on optimizing a single consensus model $\theta$ to serve the entire network:

$\min_{\theta \in \mathbb{R}^d} \left[ G(\theta) \triangleq \frac{1}{N} \sum_{i \in \mathcal{V}} \mathbb{E}_{x_i \sim P_i} l(\theta; x_i) \right]$.

In practical scenarios, the theoretical objectives of minimizing section 2.1 are achieved through empirical risk minimization (ERM) using the available local datasets. Each agent $i \in \mathcal{V}$ possesses a local dataset $D_i = \{x_{i,1}, \ldots, xi, n_i\}$. The collective dataset across all agents is denoted as $D \triangleq \bigcup_{i=1}^{N} D_i$. Therefore, the ERM problem is formulated as:

$\min_{\theta \in \mathbb{R}^d} \left[ \hat{G}_D(\theta) \triangleq \frac{1}{N} \sum_{i \in \mathcal{V}} \sum_{j=1}^{n_i} l(\theta_i; x_{i,j}) \right]$.

Decentralized learning algorithms address the consensus model optimization problem by relying solely on local agent model updates and peer-to-peer communications within the network graph (Tsitsiklis et al., 1986; Nedic & Ozdaglar, 2009). In Algorithm 1, we illustrate a typical decentralized learning process that alternates between local model updates for each agent and the integration of agents' parameters through gossip averaging with neighboring nodes based on the mixing matrix $A$.

## 2.2 METRICS

To evaluate model performance in decentralized learning, we adopt practical metrics derived from the theoretical objectives introduced earlier. Specifically, local test accuracy measures generalization on local data, while global test accuracy evaluates generalization across the entire data distribution. These two aspects are typically studied independently. Therefore, we aim to provide precise definitions for these concepts before delving into our work.

**Definition 1** (Test Accuracy). *Assuming the tasks for decentralized learning is image classification, we define the local test accuracy and global test accuracy of site $i$:*

- $\text{LocalTestAcc}_i(\theta_i) = \frac{\sum_{x_{i,j} \in D_{i\text{-th site test}}} \mathbf{1}\left(f(\theta_i, x_{i,j}) = y_{i,j}\right)}{|D_{i\text{-th site test}}|}$, *where $\mathbf{1}(\cdot)$ is the indicator function, $f(\theta_i, \cdot)$ is the model in site $i$, $x_{i,j} \in D_{i\text{-th site test}} \sim P_i$ test data on site $i$, and $y_{i,j}$ is the ground truth label for $x_{i,j}$.*

- $\text{GlobalTestAcc}(\theta_i) = \frac{\sum_{x_{i,j} \in D_{global\ test}} \mathbf{1}\left(f(\theta_i, x_{i,j}) = y_{i,j}\right)}{|D_{global\ test}|}$, *where $x_{i,j} \in D_{global\ test} \sim P$ which is the distribution of test data over all the sites.*

- $\text{ALA}(\theta_1, \cdots, \theta_n) = \frac{1}{N} \sum_i^N \text{LocalTestAcc}_i(\theta_i)$

- $\text{AGA}(\theta_1, \cdots, \theta_n) = \frac{1}{N} \sum_i^N \text{GlobalTestAcc}(\theta_i)$

- $\text{MMGA}(\theta_{merged}) = \text{GlobalTestAcc}(\theta_{merged})$

*where ALA represents **A**verage **L**ocal Test **A**ccuracy, AGA represents **A**verage **G**lobal Test **A**ccuracy, MMGA represents **M**erged **M**odel **G**lobal Test **A**ccuracy.*

In this paper, we take $\theta_{\text{merged}} = \frac{1}{N} \sum_i^N \theta_i$, which is known as model soup Wortsman et al. (2022) in the model merging community. The choice of model merging method in Decentralized Learning remains an open question, though it is not the primary focus of our work.

**Definition 2** (Gossip Gain). *We define Gossip Gain (GG) as $GG = \left( \frac{MMGA(\theta_{merged})}{AGA(\theta_1, \ldots, \theta_n)} - 1 \right) \times 100\%$*

This metric quantifies the improvement in global test accuracy achieved by merging models from all neighboring sites at a given communication step. The motivation behind introducing this metric is that, at any communication step, maximizing information exchange across all sites provides an upper bound on the achievable global generalization performance. A low Gossip Gain indicates that expanding the communication graph has limited potential to further enhance global generalization, suggesting that additional inter-site collaboration may yield diminishing returns.

## 2.3 METADATA-DEPENDENT SUPERVISED LEARNING (MDSL)

In most decentralized learning studies, researchers aim to develop algorithms that enhance knowledge transferability across different sites, improve optimization efficiency, and enhance generalization capabilities, etc. To demonstrate the effectiveness of their proposed algorithms, they typically utilize benchmark datasets such as MNIST LeCun et al. (1998), CIFAR-10, CIFAR-100 Krizhevsky (2009), or TinyImageNet Le & Yang (2015). These datasets are commonly partitioned into a predetermined number of nodes by sampling from a Dirichlet distribution, allowing for the simulation of non-IID experimental settings. The degree of non-IID data distribution is controlled by the parameter $\alpha$ in the Dirichlet distribution, where smaller values of $\alpha$ lead to a higher degree of non-IID data distribution across sites.

At each site, supervised learning is conducted using gradient descent to minimize the cross-entropy loss between the predicted probabilities and the corresponding one-hot encoded labels. While this experimental setup appears reasonable, it implicitly assumes that participants across different sites have access to shared metadata or are willing to disclose metadata information. This assumption enables each site to define classification heads by aggregating class distributions across nodes and coordinating class assignments within the classification head. We define this experimental setting as Metadata-Dependent Supervised Learning (MDSL). However, we argue that this approach contradicts the fundamental privacy-preserving principles of decentralized learning and is thus not realistic in real-world applications.

## 2.4 METADATA-AGNOSTIC ZERO-SHOT LEARNING (MAZEL)

In contrast, we propose Metadata-Agnostic Zero-Shot Learning (MAZEL), which addresses the issue of requiring different sites to share metadata and coordinate classification heads before training. In this approach, each participant independently initializes a CLIP-based model. Instead of relying on shared metadata, each site constructs its own textual template, such as "This is a photo of object", for its respective classes without disclosing this information to other sites. The participant can then store the text embedding corresponding to each class label, denoted as $T_i$.

During training, participants evaluate their models by performing inference on local test images to obtain their representations, denoted as $I$. They then compute the similarity scores between these image representations and all stored text class embeddings $T_i$. The predicted class is determined by $c^* = \arg\max_i \text{sim}(I, T_i)$.

MAZEL eliminates the need for participants to predefine a classification head that incorporates class dimensions from other sites. Consequently, it removes the necessity of exchanging metadata, providing a more realistic setting for decentralized learning compared to MDSL.

## 3 MAJOR FINDINGS AND DISCUSSIONS

In this section, we will revisit the claims that have been commonly observed in decentralized learning under the MDSL experiment settings and show that these statements are not necessarily true when switching to the MAZEL settings.

**Experiment settings** We run experiments on 16 decentralized sites, where each site trains a model under two different settings: (1) ViT-B-32 CLIP-pretrained on all parameters, ViT-B-32 CLIP-pretrained on the classification head, and ViT-B-32 ImageNet pretrained on all parameters; (2) MAZEL: ViT-B-32 CLIP-pretrained model with all parameters being trainable. Please find all results in Table B.1.

All the experiments are conducted under the setting of Random-2 De Vos et al. (2024) on 16 sites, which means that each time one other site is randomly picked to gossip. The batch size is always set to 64. Random (uniform) means that the weights of merging models are equally weighted. In Random (softmax), the weights of merging models are not equal; we calculate the scores according to Zhu et al. (2025). In the complete graph, each model communicates with everyone else each time. Local training represents that there are no communications between sites at all.

**Dataset** To compare MDSL and MAZEL, we require a setting where MDSL can be meaningfully implemented. Since MDSL is not feasible in scenarios where different nodes possess distinct datasets, we simulate non-IID data distributions across sites by sampling from a Dirichlet distribution with $\alpha = 0.1$. We conduct our experiments using the CIFAR-100 Krizhevsky (2009) and the Kvasir V2 Pogorelov et al. (2017a). The dataset assigned to each site are biased samples drawn from these datasets. CIFAR-100 is a widely used dataset, and it is highly probable that similar images were included in the CLIP model pretraining data . To mitigate this potential overlap, we also evaluate our approach on Kvasir V2, a medical imaging dataset that is unlikely to be part of CLIP's pretraining dataset.

This hypothesis is supported by the performance: it achieves about 53% accuracy on the CIFAR-100 test, whereas its performance on the Kvasir V2 test set is 0% indicating Kvasir V2 were not part of pretraining data.

**Key observation 1**: *Under MAZEL, local models typically would not fail to generalize well to a global test set under highly heterogeneous scenarios.*

As shown by the results in Table B.1, Claim 1, that local models generally fail to generalize well to a global test set in highly heterogeneous scenarios, holds true in the MDSL setting. AGA, which quantifies the global generalization performance of local models, consistently demonstrates poor performance across all MDSL configurations. However, this trend does not persist in the MAZEL setting, where local models exhibit improved generalization capabilities.

In MDSL, the best-performing configuration involves using a CLIP-pretrained model with only the classification head unfrozen during training. In the CIFAR-100 experiments, we observe that AGA improves from 33.70% (local training) to 45.52% when adopting random uniform decentralized learning, yielding an 11.82% enhancement in the global generalization performance of local models. In contrast, under MAZEL, AGA increases from 52.84% to 80.65%, demonstrating a substantial improvement. This suggests that local models in MAZEL inherently possess strong global generalization capabilities.

A similar trend is observed in the Kvasir V2 experiments, where the improvement is even more pronounced. The AGA increases by 40.49%, reaching 91.99%, which is remarkably close to the ALA of 93.54%.

One potential source of confusion is that both MDSL and MAZEL exhibit better performance on Kvasir V2 than on CIFAR-100, despite the fact that Kvasir V2 being dissimilar to CLIP's pretraining data. We attribute this discrepancy to the difference in class cardinality: Kvasir V2 contains only 8 classes, meaning that even a random guess yields a 12.5% accuracy, whereas CIFAR-100 has 100 classes, making it a significantly more challenging classification task.

**Key observation 2**: *Under MAZEL, local models converge faster in terms of ALA, AGA and GG.* Previous studies Lian et al. (2017); Koloskova et al. (2020); Kong et al. (2021) have shown that decentralized learning under MDSL exhibits slow convergence. Our findings align with these results: neither the CLIP-pretrained nor the ImageNet-pretrained model converges within 8000 training steps under MDSL full finetuning. Even when finetuning only the classification head of the CLIP-pretrained model, convergence still requires approximately 2x steps compared to the MAZEL setting.

Furthermore, under the MAZEL setting, the Gossip Gain (GG) score remains below 4% for both Random (Uniform) and Random (Influence-Weighted) strategies. In contrast, under MDSL, the GG score ranges from 21.06% to 72.02%, indicating significant potential for further improving the global generalization performance of local models. This suggests that, even when the ALA and AGA curves have reached a plateau, MDSL still has potential for enhancement through more gossip.

In conclusion, the above results highlight that decentralized learning under MDSL and MAZEL can yield significantly different outcomes in key metrics, such as the global generalization of local

models and convergence speed. Given that MAZEL better aligns with real-world decentralized learning applications, we strongly encourage researchers to conduct experiments within the MAZEL framework to ensure a more comprehensive evaluation of proposed algorithms.

# 4 MAZEL BASELINES: 8-SITE AND 16-SITE BENCHMARKS

To advance decentralized learning within the MAZEL framework, we introduce two benchmark baselines. They are (1) 8-Site Baseline: Each site utilizes a distinct real-world dataset: MNIST LeCun (1998), Cars Krause et al. (2013), DTD Cimpoi et al. (2014), EuroSAT Helber et al. (2019), GTSRB Stallkamp et al. (2011), RESISC45 Cheng et al. (2017), SUN397 Xiao et al. (2016), SHVN Netzer et al. (2011). (2) 16-Site Baseline: Each site is assigned a different real-world dataset: MNIST, Cars, DTD, EuroSAT, GTSRB, RESISC45, SUN397, SVHN, Dogs Khosla et al. (2011), CUB-200-2011 Wah et al. (2011), Weather Xiao et al. (2021), MangoLeafBD Ahmed et al. (2023), Garbage CCHANG (2018), Beans Lab (2020), Kvasir Pogorelov et al. (2017b), and FashionMNIST Xiao et al. (2017).

For the 8-site baseline, we selected these specific datasets because they have been widely adopted in the model merging community as standard benchmarks. Model merging plays a critical role in decentralized learning, as it occurs at every gossip communication step. By aligning our dataset selection with those commonly used in the model merging community, we aim to bridge research efforts between the two fields. This alignment enables decentralized learning researchers to effectively evaluate different model merging techniques and determine which approach best enhances the performance of gossip-based updates.

However, experiments with only 8 sites may not fully capture the complexities of decentralized learning. Therefore, we expanded the benchmark to 16 sites, incorporating additional datasets inspired by those used in the EMR-merging study Huang et al. (2024). This extension ensures a more diverse and comprehensive evaluation, enabling researchers to analyze decentralized learning performance across a broader range of heterogeneous data distributions.

Below, we present experimental results evaluating various techniques on the MAZEL Baseline 8-Site in Table B.2 and MAZEL Baseline 16-Site benchmarks in Table A.1.

We implemented two gossip schedulers to regulate the timing of communication rounds: the Two-Stage Gossip Scheduler and the Cosine Gossip Scheduler.

From our experiments, we observed that allocating more communication rounds to the early training stage rather than the later stages is beneficial. Excessive communication toward the end of training may disrupt model convergence.

**Two-Stage Gossip Scheduler**    The Two-Stage Gossip Scheduler divides the training process into two phases: the early stage and the convergence stage, each with a distinct gossip interval hyperparameter. The early stage involves more frequent communication, while the convergence stage adopts a less frequent communication schedule to prevent unnecessary perturbations. However, even in the later stage, where the gossip interval is larger, careful hyperparameter tuning is required to ensure that no gossip occurs in the final training steps, as this could destabilize the model.

**Cosine Gossip Scheduler**    To further address this issue, we introduce the Cosine Gossip Scheduler, inspired by the cosine learning rate scheduler Loshchilov & Hutter (2016). This approach gradually reduces the probability of gossiping as training progresses, ensuring that the communication frequency is significantly lower toward the end of training. This minimizes potential disruptions in the final optimization steps, preventing oscillations in model performance.

In conclusion, the local training step and the gossip step in decentralized learning function like a non-zero-sum tug-of-war. The local training step promotes local models' ability to generalize locally, whereas the gossip step enhances their global generalization. Increasing the learning rate strengthens local training, thereby benefiting ALA, while more frequent gossip updates improve AGA. As shown in Table B.2, the combination of influence-based weighting, a uniform learning rate scheduler with warmup, and a frequent cosine gossip scheduler leads to higher ALA and AGA while maintaining a low GG.

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

# A   ADDITIONAL EXPERIMENT RESULTS

Table A.1: Comparison of various communication frequency, learning rate schedulers, and gossip schedulers. All experiments using random neighbor selection with influence-based weights Zhu et al. (2025) using CLIP-pretrained ViT-B/32 on 16 datasets. Experimented configurations include: **A**: cosine gossip scheduler (9 total gossips), **B**: cosine gossip scheduler (15 total gossips), **C**: cosine gossip scheduler (46 total gossips), **D**: Cosine learning rate scheduler, **E**: uniform learning rate scheduler with warmup, **F**: two-stage gossip scheduler (9 total gossips).

|              | D+F   | A+D   | A+E   | B+E   | C+E   |
|--------------|-------|-------|-------|-------|-------|
| ALA          | 63.17 | 81.81 | 84.18 | 83.66 | 83.09 |
| AGA          | 58.16 | 57.64 | 55.78 | 59.77 | 70.34 |
| MMGA         | 58.77 | 61.03 | 60.96 | 63.57 | 73.19 |
| (ALA+AGA)/2  | 60.67 | 69.73 | 69.98 | 71.72 | 76.72 |
| Gossip Gain  | 1.05% | 5.88% | 9.29% | 6.36% | 4.05% |

# B   RELATED WORK

**Decentralized Learning**. Decentralized learning has emerged as a powerful paradigm for distributed optimization, enabling collaborative model training across multiple nodes without the need for a centralized coordinator. Decentralized Stochastic Gradient Descent (DSGD) (Lian et al., 2017) serves as a prominent example of decentralized learning algorithms. Building on the principles of DSGD, the field of decentralized learning has expanded rapidly, driven by the need for adaptable and efficient solutions in diverse and dynamic environments. Modern decentralized algorithms have evolved to address challenges such as time-varying network topologies (Nedi'c & Olshevsky, 2014; Lu & Wu, 2020; Koloskova et al., 2020; Ying et al., 2021), enabling robust performance even in scenarios where communication links between nodes fluctuate. Moreover, the incorporation of asynchronous communication protocols (Lian et al., 2018; Xu et al., 2021; Nadiradze et al., 2021; Bornstein et al., 2023) has empowered decentralized methods to overcome latency and synchronization barriers, further enhancing their scalability. Another critical advancement is the ability to handle heterogeneous data distributions (Tang et al., 2018; Vogels et al., 2021; Le Bars et al., 2023), which mirrors the realities of non-IID data commonly encountered in real-world decentralized systems.

**Zero-shot Classification**. CLIP Radford et al. (2021a) pioneers contrastive learning between images and text, demonstrating that large-scale natural language supervision enables strong zero-shot transfer across diverse vision tasks. SigLIP Zhai et al. (2023) introduces a pairwise sigmoid loss for language-image pretraining, enabling efficient scaling of batch sizes while improving zero-shot accuracy on ImageNet. CLAP Wu et al. (2023) extends contrastive learning to the audio domain, training a large-scale language-audio model using feature fusion and keyword-to-caption augmentation for superior zero-shot classification and retrieval. DINOv2 Oquab et al. (2023) advances self-supervised learning by training large ViT models on curated datasets, producing robust all-purpose visual features that surpass OpenCLIP in most benchmarks.

**Model-merging** "Model soups"Wortsman et al. (2022) averages the weights of multiple fine-tuned models improves accuracy and robustness without increasing inference time. Task vectors Ilharco et al. (2023) represent directions in weight space; by adding or subtracting these vectors, models can acquire or diminish specific capabilities. DARE Yu et al. (2023) demonstrated that language models could absorb new abilities by assimilating parameters from homologous models without retraining, a process facilitated by the DARE method to sparsify delta parameters. Ties-merging Yadav et al. (2023) addressed parameter interference in model merging, a method that resolves conflicts by resetting minimally changed parameters and aligning parameter signs. AdaMerging Yang et al. (2024) is an adaptive approach that autonomously learns merging coefficients without relying on original training data, enhancing performance across multiple tasks. MAP Li et al. (2024), a low-compute algorithm that efficiently identifies a Pareto set of scaling coefficients for merging models, reflecting the trade-offs involved. Collectively, these studies contribute to the evolving landscape of model merging, offering diverse strategies to combine models effectively.

## B 1    ALGORITHM FOR DECENTRALIZED LEARNING

---

**Algorithm 1** DECENTRALIZED LEARNING

---

1: **input** For each node $i \in \mathcal{V}$, initialize $\theta_i^0 \in \mathbb{R}^d$, number of iterations $T$, mixing matrix $A$
2: **for** $t = 0, \ldots, T$ **do**
3:     **for** $i \in \mathcal{V}$ in parallel **do**
4:         Sample training data batch $x_{i,j}^t$ from $P_i$,
        $\theta_i^{t+\frac{1}{2}} = \text{Optimizer}(\theta_i^t; x_{i,j}^t)$                                        $\triangleright$ Local training
5:         Send parameters $\theta_i^{t+\frac{1}{2}}$ to out-neighbor(s) and receive $\{w_l^{t+\frac{1}{2}}\}_{l \in \mathcal{N}_{\text{in}}(i)}$ from in-neighbor(s) $\triangleright$ Communication
6:         $\theta_i^{t+1} = \sum_{l \in \mathcal{N}_{\text{in}}(i)} A_{i,l} w_l^{t+\frac{1}{2}}$                          $\triangleright$ Gossip averaging
7:     **end for**
8: **end for**

---

## B 2    THE IMPACT OF GOSSIP ROUNDS ON LOCAL GENERALIZATION AND GLOBAL GENERALIZATION

**Experiments settings**    We conduct experiments on 16 decentralized sites, where each site trains a model under two different settings: (1) MDSL: ViT-B-32 CLIP-pretrained model with only the classification head being trainable; (2) MAZEL: ViT-B-32 CLIP-pretrained model with all parameters being trainable.

All experiments follow the Random-2 (Uniform) protocol, meaning that at each gossip round, a site randomly selects one other site for communication, and models are merged using equal weights.

For MDSL, training consists of 8050 steps with gossip intervals set at 100, 200, 300, 400, 500, 600, 1000, 2000, 3000, and 6000 steps. The corresponding number of gossip rounds is 80, 40, 26, 20, 16, 13, 8, 4, 2, and 1, respectively.

Since our previous results in table B.1 indicate that convergence is faster under MAZEL, we set the total number of training steps for MAZEL to be 2150. The gossip intervals are set at 50, 100, 200, 300, 400, 500, 1000, and 2000, with the corresponding number of gossip rounds being 43, 21, 10, 7, 5, 4, 2, and 1, respectively.

**Dataset**    We follow the same data partitioning method as in **??**, distributing CIFAR-100 among the 16 sites.

**Results analysis**    As illustrated in Figure B.1, in MAZEL, Average Local Accuracy (ALA) initially decreases as the number of communication rounds increases but subsequently improves. In contrast, Average Global Accuracy (AGA) consistently increases throughout. This observation suggests that for practitioners who prioritize local generalization, there exists a critical range of communication rounds that should be avoided. Specifically, when the total communication round is near 5, local generalization is at its lowest. Either increasing or decreasing the gossip frequency improves ALA, leading to better local model performance.

However, when considering AGA, global generalization continuously improves as the number of communication rounds increases. When averaging ALA and AGA, the highest performance is observed at a gossip interval of 200 steps. Nevertheless, this should not be interpreted as a universal optimal setting, as the ideal number of communication rounds depends on the priorities and preferences of the participating site owners. Moreover, this U-shaped pattern becomes even more evident in subsequent experiments (Figure B.1).

In contrast, under MDSL, both ALA and AGA exhibit a nearly monotonous increasing trend as the number of communication rounds increases.

## B 3    ADDITIONAL FIGURES AND TABLES

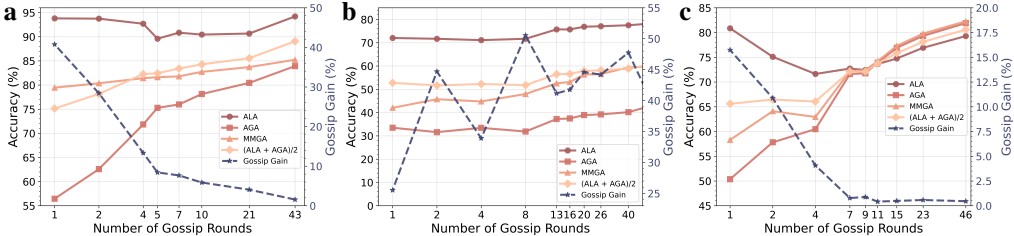

Figure B.1: Impact of the number of gossip rounds on various performance metrics for CLIP-pretrained ViT-B/32 models Radford et al. (2021b) across different finetuning strategies and dataset settings. Abbreviations: **ALA**: Average local test accuracy, **AGA**: Average global test accuracy, **MMGA**: Merged model global test accuracy, **MDSL**: Metadata-dependent supervised learning, **MAZEL**: Metadata-agnostic zero-shot learning. **Subfigure a**: Full finetuned under MAZEL settings using the CIFAR-100 dataset. **Subfigure b**: Classification head-only finetuned under MDSL settings using the CIFAR-100 dataset. **Subfigure c**: Fully finetuned under MAZEL settings using eight diverse datasets described in section 4.

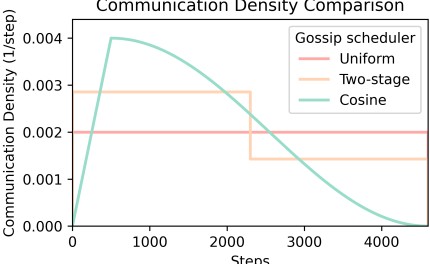

Figure B.2: Comparison of different gossip schedulers over 4,600 training steps, all with 9 total gossip rounds. The uniform gossip scheduler has fixed intervals of 500 steps. The two-stage scheduler uses a higher frequency in the first half, followed by lower frequency in the second half. The cosine gossip scheduler uses a cosine decay with an initial warmup.

Table B.1: Comparison of decentralized learning performance under MDSL and MAZEL across different datasets and finetuning strategies. Abbreviations: **ALA**: Average local test accuracy, **AGA**: Average global test accuracy, **MMGA**: Merged model global test accuracy, **MDSL**: Metadata-dependent supervised learning setting, **MAZEL**: Metadata-agnostic zero-shot learning setting. Experiments were conducted using two datasets: CIFAR-100 and Kvasir v2. We evaluate CLIP-pretrained Radford et al. (2021b) and ImageNet-pretrained Dosovitskiy et al. (2020) ViT-B/32 models under two finetuning strategies: **Full FT** (full finetuning) and **Classification head FT** (classification head-only finetuning). We include four learning topologies: random communication with uniform weights, random communication with influence-based weights Zhu et al. (2025), complete communication graph with uniform weights, and local training only. For the MDSL setting, training consists of 8,000 steps with a gossip interval of 100 steps, while for the MAZEL setting, training consists of 2,050 steps with the same gossip interval.

| | | CIFAR-100 | | | | Kvasir V2 | |
| | | MDSL | | | MAZEL | MDSL | MAZEL |
| | | CLIP pretrained | | ImageNet pretrained | CLIP pretrained | CLIP pretrained | CLIP pretrained |
| | | Full FT | Classification head FT | Full FT | Full FT | Classification head FT | Full FT |
|---|---|---|---|---|---|---|---|
| Random (Uniform) | ALA | 30.42 | 78.95 | 38.56 | 90.67 | 88.17 | 93.54 |
| | AGA | 7.29 | 45.52 | 23.76 | 80.65 | 65.1 | 91.99 |
| | MMGA | 12.54 | 60.31 | 29.15 | 83.72 | 83.33 | 93.75 |
| | (ALA+AGA)/2 | 18.86 | 62.24 | 31.16 | 85.66 | 76.64 | 92.77 |
| | Gossip Gain | 72.02% | 32.49% | 22.69% | 3.81% | 28.00% | 1.91% |
| | Converge Steps | >8000 | 5860 | >8000 | 1100 | 6700 | 3450 |
| Random (Influence-weighted) | ALA | 33.32 | 80.39 | 39.31 | 92.47 | 84.02 | 93.91 |
| | AGA | 8.10 | 47.40 | 23.17 | 81.06 | 68.10 | 92.47 |
| | MMGA | 12.71 | 61.88 | 28.05 | 83.87 | 84.90 | 94.21 |
| | (ALA+AGA)/2 | 20.71 | 63.90 | 31.24 | 86.77 | 76.06 | 93.19 |
| | Gossip Gain | 56.91% | 30.55% | 21.06% | 3.47% | 24.67% | 1.88% |
| | Converge Steps | >8000 | 7200 | >8000 | 1100 | 7100 | 3600 |
| Complete (Uniform)) | ALA | 11.93 | 71.22 | 28.83 | 87.09 | 91.68 | 94.56 |
| | AGA | 7.81 | 62.83 | 29.37 | 84.88 | 90.63 | 94.27 |
| | MMGA | 7.70 | 62.93 | 29.34 | 84.92 | 91.15 | 94.27 |
| | (ALA+AGA)/2 | 9.87 | 67.03 | 29.10 | 85.99 | 91.16 | 94.42 |
| | Gossip Gain | -1.41% | 0.16% | -0.10% | 0.05% | 0.57% | 0.00% |
| | Converge Steps | > 8000 | 7200 | >8000 | 4350 | 4400 | 1250 |
| Local training | ALA | 33.48 | 90.61 | 40.43 | 93.30 | 94.11 | 97.65 |
| | AGA | 2.88 | 33.70 | 12.98 | 52.84 | 33.91 | 51.50 |
| | MMGA | 1.00 | 44.25 | 19.77 | 79.31 | 36.23 | 68.75 |
| | (ALA+AGA)/2 | 18.18 | 62.16 | 26.71 | 73.07 | 59.01 | 74.58 |
| | Gossip Gain | -65.28% | 31.31% | 52.31% | 50.09% | 51.53% | 33.50% |
| | Converge Steps | > 8000 | 3300 | >8000 | 2100 | 4600 | 2400 |

Table B.2: Comparison of various communication frequencies, learning rate schedulers, and gossip schedulers under the proposed MAZEL setting. Abbreviations: **ALA**: Average local test accuracy, **AGA**: Average global test accuracy, **MMGA**: Merged model global test accuracy, **MAZEL**: Metadata-agnostic zero-shot learning, **Local**: Local training only with no gossip. All experiments used CLIP-pretrained ViT-B/32 Radford et al. (2021b) on the 8 datasets described in section 4. Experimented configurations include: **A**: Random neighbor selection with influence-based weights Zhu et al. (2025), **B**: Random neighbor selection with uniform weights, **C**: Cosine learning rate scheduler, **D**: Uniform learning rate scheduler with warmup, **E**: Two-stage gossip scheduler (9 total gossip rounds), **F**: Cosine gossip scheduler (9 total gossip rounds), **G**: Uniform gossip scheduler (9 total gossip rounds), **H**: Cosine gossip scheduler (15 total gossip rounds), **I**: Cosine gossip scheduler (46 total gossip rounds).

| | A+D+F | B+D+F | A+C+E | B+C+E | A+D+G | A+D+E | A+D+H | A+D+I | A+C+F | Local |
|---|---|---|---|---|---|---|---|---|---|---|
| ALA | 72.62 | 72.38 | 76.81 | 74.56 | 74.57 | 80.68 | 76.82 | 80.84 | 73.52 | 88.66 |
| AGA | 73.92 | 70.55 | 76.86 | 71.81 | 76.18 | 75.44 | 79.05 | 82.95 | 75.67 | 38.29 |
| MMGA | 74.31 | 71.00 | 77.73 | 73.44 | 76.61 | 78.32 | 79.35 | 83.25 | 76.03 | 58.10 |
| (ALA+AGA)/2 | 73.27 | 71.47 | 76.84 | 73.19 | 75.38 | 78.06 | 77.94 | 81.90 | 74.60 | 63.48 |
| Gossip Gain | 0.53% | 0.64% | 1.13% | 2.27% | 0.56% | 3.82% | 0.38% | 0.36% | 0.48% | 51.74% |

