# OpenReview forum: "Rethinking Decentralized Learning: Towards More Realistic Evaluations with a Metadata-Agnostic Approach"
_ICLR.cc/2025/Workshop/MCDC — MCDC @ ICLR 2025_

### Official Review · Reviewer_sb3X · 2025-02-25

**Rating:** 4
**Confidence:** 5
**Fit:** 4

**Summary:**

This paper challenges the metadata awareness used in decentralized learning evaluation, i.e., simulating various non-IID degrees of data distribution through Dirichlet functions. The authors argue that such an evaluation tactic assumes access to the total number of classes as shared global information, breaking privacy constraints. To address this challenge, a new experimental setting called MAZEL (Metadata-Agnostic Zero-Shot Learning) is proposed. This setting relies on image/dataset captioning at each client through a CLIP-based model. The similarity between the predicted embedding and stored class embeddings determines the predicted class.

**Reason For Giving A Higher Score:**

--

**Reason For Giving A Lower Score:**

1. The motivation is not well-grounded.

2. The results need to be explained better through some qualitative analysis or arguments.

**Strengths And Weaknesses:**

Strengths:

1. The paper is well-structured and the literature review is quite thorough.

2. Evaluation is done across a range of datasets.

Weaknesses:

1. My major issue with this paper is the fundamental assumption in decentralized/federated learning the authors challenge. The degree of non-IID data (represented by $\alpha$ in most papers) is used for evaluation, but this technique is in no way part of the training loop. For example, state-of-the-art decentralized algorithms tackling data heterogeneity [1-3] do not base their approach on the degree of heterogeneity. $\alpha$ is an evaluation tactic, and even if one is not aware of it, it is possible to evaluate but this just provides a finer control and structured evaluation. I do not think that this breaks data privacy.

2. The results show that the technique proposed in this paper, MAZEL, performs better than MDSL. However, the reasoning behind it is lacking, and the reader is left to figure that out themselves. After spending some time, I think it performs better because embeddings are a form of soft label compared to the hard labels used in the traditional MDSL setup. I may be wrong, but again, the paper doesn't help me understand this.

**Suggestions:**

Overall, I believe the motivation is not convincing, but this technique can be explored in its own light, given how well it performs. I would strongly encourage the authors to spend some time explaining their results better for future submissions to any other venues.

---

### Official Review · Reviewer_Edmg · 2025-02-27

**Rating:** 7
**Confidence:** 3
**Fit:** 3

**Summary:**

This paper challenges current experimental approaches in decentralized learning research by highlighting a discrepancy between research settings and real-world constraints. The authors identify two distinct paradigms:

Metadata-Dependent Supervised Learning (MDSL): The conventional approach where participants share metadata (like class labels) across sites, typically using datasets like CIFAR-100 partitioned with Dirichlet distributions.
Metadata-Agnostic Zero-Shot Learning (MAZEL): A proposed approach that better reflects real-world privacy constraints, where participants cannot share metadata across sites. Under MAZEL, local models generalize well to global test sets, contradicting conventional wisdom from MDSL settings. MAZEL settings show faster convergence in terms of Average Local Accuracy (ALA), Average Global Accuracy (AGA), and Gossip Gain compared to MDSL. Different gossip scheduling strategies significantly impact model performance, with early communication being more beneficial than later-stage communication. The authors argue that MAZEL provides a more realistic evaluation framework for decentralized learning that better aligns with real-world privacy constraints, and they encourage researchers to adopt this framework for future evaluations.

**Reason For Giving A Higher Score:**

- The paper introduces a significant paradigm shift in evaluating decentralized learning systems, addressing a genuine gap between research practices and real-world constraints.
- The paper goes beyond criticism by offering concrete solutions (MAZEL framework and gossip schedulers) that can improve decentralized learning in practice.
- The experimental design is thorough and considers multiple models, datasets, and training configurations to support their claims.

**Reason For Giving A Lower Score:**

- The paper relies heavily on empirical results without providing adequate theoretical explanations for why MAZEL outperforms MDSL in key metrics.
- While the paper presents a new evaluation framework, the technical contributions (gossip schedulers) could be viewed as incremental.
- The experiments are limited to visual classification tasks with CLIP models and don't demonstrate broader applicability to other domains or model types.

**Strengths And Weaknesses:**

Strengths:

- The authors provide extensive experiments across multiple datasets, models, and settings to support their claims, comparing MDSL and MAZEL approaches directly.
- The proposed gossip schedulers (Two-Stage and Cosine) offer concrete solutions to optimize communication efficiency in decentralized learning scenarios.

Weaknesses
- While the empirical results are strong, the paper lacks deeper theoretical analysis explaining why MAZEL settings lead to better generalization and faster convergence than MDSL.

**Suggestions:**

Expand the Theoretical Foundation: Develop a theoretical framework that explains why MAZEL settings lead to better generalization and faster convergence than MDSL. This would strengthen your empirical findings and provide deeper insights for the research community.
Scale the Experiments: Test your approach on larger networks (e.g., 50+ sites) to demonstrate scalability.

---

### Official Review · Reviewer_p49a · 2025-03-01

**Rating:** 8
**Confidence:** 3
**Fit:** 4

**Summary:**

The paper critiques current decentralized learning evaluations that assume shared metadata and introduces Metadata-Agnostic Zero-Shot Learning (MAZEL), a more realistic setting where nodes lack metadata synchronization. Empirical results show that long-standing claims about poor generalization and slow convergence do not hold under MAZEL. The paper benchmarks decentralized methods on 8–16 diverse datasets and proposes new gossip schedulers to improve communication efficiency.

**Reason For Giving A Higher Score:**

Well-written paper with a clear motivation and showcase depth of understanding.

**Reason For Giving A Lower Score:**

.

**Strengths And Weaknesses:**

Strength:
1. Introduces a realistic, privacy-preserving decentralized learning framework.
2. Provides empirical benchmarks with diverse datasets and new communication strategies.
3. Challenges existing assumptions and shows that decentralized models generalize well under MAZEL.

Weaknesses:
1. Relies on CLIP embeddings, which may limit applicability to non-image tasks.
2. Computational overhead of metadata-agnostic approaches is not fully analyzed.

**Suggestions:**

Refer to weaknesses

---

### Decision · Program_Chairs · 2025-03-06

**Decision:**

Accept

**Comment:**

The paper critiques current decentralized learning evaluations that assume shared metadata by highlighting a discrepancy between research settings and real-world constraints. The paper recieved scores with high variance. We suggest the authors to  incorporate the comments and suggestions from reviewer sb3X to strengthen the paper. The paper seems relevant to the topic of decentralized training. Overall, we're recommend to accept this work to the workshop.